# A Study on the Time–Effect and Dose–Effect Relationships of Polysaccharide from *Opuntia dillenii* against Cadmium-Induced Liver Injury in Mice

**DOI:** 10.3390/foods11091340

**Published:** 2022-05-04

**Authors:** Ting Liu, Bianli Li, Xin Zhou, Huaguo Chen

**Affiliations:** 1Key Laboratory for Information System of Mountainous Areas and Protection of Ecological Environment, Guizhou Normal University, 116 Baoshan North Road, Guiyang 550001, China; tingliu@gznu.edu.cn (T.L.); bianlili@gznu.edu.cn (B.L.); zhouxin@gznu.edu.cn (X.Z.); 2Guizhou Engineering Laboratory for Quality Control & Evaluation Technology of Medicine, Guizhou Normal University, 116 Baoshan North Road, Guiyang 550001, China

**Keywords:** *Opuntia dillenii* (Ker-Gaw) Haw. polysaccharide, cadmium-induced liver injury, time–effect relationship, dose–effect relationship

## Abstract

The purpose of this study was to evaluate the protective effect of *Opuntia dillenii* (Ker-Gaw) Haw. polysaccharide (ODP) against cadmium-induced liver injury. Cadmium chloride (CdCl_2_) was used to construct a mice evaluation model, and the indicators chosen included general signs, liver index, biochemical indicators, blood indicators, and pathological changes. A dose of 200 mg/kg ODP was applied to the mice exposed to cadmium for different lengths of time (7, 14, 21, 28, and 35 days). The results showed that CdCl_2_ intervention led to slow weight growth (reduced by 13–20%); liver enlargement; significantly increased aspartate aminotransferase (AST, 45.6–52.0%), alanine aminotransferase (ALT, 26.6–31.3%), and alkaline phosphatase (ALP, 38.2–43.1%) levels; and significantly decreased hemoglobin (HGB, 13.1–15.2%), mean corpuscular hemoglobin (MCH, 16.5–19.3%), and mean corpuscular hemoglobin concentrations (MCHC, 8.0–12.7%) (*p* < 0.01). In addition, it led to pathological features such as liver cell swelling, nuclear exposure, central venous congestion, apoptosis, and inflammatory cell infiltration. The onset of ODP anti-cadmium-induced liver injury occurred within 7 days after administration, and the efficacy reached the highest level after continuous administration for 14 days, a trend that could continue until 35 days. Different doses (50, 100, 200, 400, and 600 mg/kg) of ODP have a certain degree of protective effect on cadmium-induced liver injury, showing a good dose–effect relationship. After 28 days of administration of a 200 mg/kg dose, all pathological indicators were close to normal values. These findings indicated that ODP had positive activity against cadmium-induced liver injury and excellent potential for use as a health food or therapeutic drug.

## 1. Introduction

Cadmium pollution is an international environmental problem seriously endangering human health [1]. The United States Agency for Poison and Disease Registry has listed cadmium as the seventh most dangerous substance to human health [2]. Cadmium in the environment mainly comes from the Earth’s crust and industrial pollution [3]. In industrial production, cadmium is usually discharged into the environment through wastewater and waste gas [4], polluting water and soil, then polluting crops through irrigation and planting, before finally accumulating in the human body through food and water, which seriously damages human health [3].

The liver is one of the main targets of cadmium exposure, and the accumulation of cadmium in the liver can cause abnormal liver function [5]. However, a great deal of research has been carried out on this issue, and exciting results have been achieved in the treatment of cadmium-induced liver injury [6]. Some traditional liver disease drugs, such as γ-glutamyl cysteine [7] and glutathione [8], can alleviate the damage to liver cells and reduce the content of cadmium in the liver. Cardiovascular and inflammatory disease drugs can help to resist cadmium acute liver injury but aggravate renal injury [9]. Piroxicam, a nonsteroidal anti-inflammatory drug, is widely used in the treatment of cadmium-induced liver injury, but it has certain side effects [10]. In general, although chemical drugs have a certain therapeutic effect on liver injury, most have shortcomings such as toxicity and harmful side effects [11], which inevitably cause secondary damage to the body during the course of treatment. In recent years, researchers have attempted to find antidotes [12] derived from natural products. For example, glycyrrhiza and salidroside can help to prevent cadmium-induced cell death [13,14], and Ganoderma triterpenoids [15] can reduce the accumulation of cadmium in the liver and protect liver cells from damage. Moreover, most plant polysaccharides are relatively nontoxic and do not cause significant side effects [16]. Many polysaccharides and their derivatives have been used in various medical applications and have shown promise in the treatment of liver disease [17].

*Opuntia dillenii* (Ker-Gaw) Haw. has a long history as a medicinal plant in China and has also been used in various fields, such as in food, animal feed, and as bonsai [18]. In recent years, with the progress of technology and the increasing demand for health products, the nutritional and pharmacological potential of *O. dillenii* has received increasing attention from researchers [19]. The physiological function of *O. dillenii* is closely related to its bioactive components, and polysaccharides are considered to be some of its most important bioactive components [20]. Although the liver-protective effects of *O. dillenii* have been thoroughly explored in animal models [21], there is currently a lack of studies on the role of *O. dillenii* in liver injury caused by the heavy metal cadmium. In our previous studies [22], processes for the extraction and purification of ODP were systematically investigated, and it was confirmed that ODP has good antioxidant activity. ODP also has a variety of biological activities such as antitumor and neuroprotective activities [14,23]. In addition, cactus fruit extract has an obvious therapeutic effect on histopathological changes in the liver of mice exposed to Cd [24]. *O. dillenii* extract is nontoxic via the oral route and appears to be non-cyto-, hepato-, nephro-, or genotoxic, thereby supporting its use against a variety of ailments [25]. Due to its good biological activity and safety, ODP may have potentially protective and therapeutic effects against cadmium-induced liver injury. In order to verify this claim, in this study, we used mice as experimental animals to construct a cadmium-induced liver injury model based on CdCl_2_. Based on this, the question of whether ODP has protective effects on cadmium-induced liver injury was explored from the perspectives of its time–effect and dose–effect relationships. Furthermore, the starting time of its effect, optimal effect range, and whether it has a dose–effect relationship were clarified.

## 2. Materials and Methods

### 2.1. Materials and Reagent

ODP (the purity was ≥95%) was obtained from our lab. Cadmium chloride (CdCl_2_) was purchased from Tianjin Zhiyuan Chemical Reagent Co., Ltd. Bifendate pills (batch number: 19J200601) were purchased from Wanbangde Pharmaceutical Group Co., Ltd. Test kits for alanine transaminase (ALT), aspartate transaminase (AST), and alkaline phosphatase (ALP) were obtained from Shanghai Guduo Biotechnology Co., Ltd, China. Additionally, all the other reagents used were of analytical grade.

### 2.2. Animals

SPF male KM mice (body weight 18–22 g) were purchased from Changsha Tianqin Biotechnology Co., LTD. (Changsha, China), and the animal license was No. SCXK (Xiang) 2019-0014. Mice were fed at a temperature of 22 ± 0.5 ℃, a humidity of 55 ± 5%, and a light cycle of 12 h:12 h.

### 2.3. Ethical Approval

All animal experiments in this study were carried out according to the Laboratory Animal Care Guide 8th Edition (2011) [26] and approved by the Animal Care and Use Committee of Guizhou Normal University. The project identification code was GZNU20210206, and the approval date was 10 February 2021.

### 2.4. Experiment Design

#### 2.4.1. Establishment and Evaluation of Cadmium-Induced Liver Injury Model

An appropriate amount of CdCl_2_ was precisely weighed out and dissolved in pure water to prepare a 0.2 mg/mL solution. A total of 20 male KM mice were randomly divided into two groups after 7 days of adaptive feeding—namely, into a normal control group (NC) and a CdCl_2_ treatment group (MC). Mice in the MC group were intraperitoneally injected with 0.2 mL of CdCl_2_ solution every day for 6 days a week and treated continuously for 3 weeks [27,28]. NC Group mice were treated according to the MC Group’s treatment program, and the injection solution administered was normal saline. During the experiment, the appetite, hair, autonomous activity, and mental state of the mice were observed every day, and their body weight was weighed once every 3 days. Mice were sacrificed on day 21, and blood and liver samples were collected for subsequent analyses.

#### 2.4.2. Time–Effect Relationship Exploration

In this section, appropriate amounts of bifendate and ODP were precisely weighed out and added to pure water to prepare 21 and 20 mg/mL solutions, respectively. Then, 5-time nodes of 7 days, 14 days, 21 days, 28 days, and 35 days were set for observation. In the pharmacodynamic study of each time node, 40 mice were randomly divided into 4 groups—namely, an NC, an MC, a bifendate-treated group (YC), and an ODP-treated group (ODPC). Mice in the MC, YC, and ODPC groups were intraperitoneally injected with 0.2 mL of CdCl_2_ solution every day for 6 days a week, while the NC group mice were intraperitoneally injected with the same amount of normal saline. Then, the mice in the YC and ODPC groups were given bifendate (21 mg/mL) and ODP (20 mg/mL) solutions by intragastric administration at a dose of 0.1 mL per 10 g body weight, while the mice in the NC and MC groups were given equal volumes of normal saline once a day. During the experiment, the appetite, hair, autonomous activity, and mental state of the mice were observed every day, and their body weight was weighed once every 3 days. Mice at each research time node were sacrificed, and blood and liver samples were collected for subsequent analysis.

#### 2.4.3. Dose–Effect Relationship Exploration

Eighty mice were randomly divided into 8 groups: groups 1 to 3 were given NC, MC, and YC, respectively, and groups 4 to 8 were given ODP (50, 100, 200, 400, 600 mg/kg) at different doses. Mice in groups 2 to 8 were intraperitoneally injected with 2 mg of CdCl_2_ per kg body weight, and mice in group 1 were injected with an equal volume of normal saline once every three days. Then, the YC group was intragastrically administrated 180 mg/kg of glutathione suspension; the ODP groups were given 50, 100, 200, 400, and 600 mg/kg of ODP solution; and the NC and MC groups were administrated equal volumes of normal saline. Mice in each group were treated once a day for 28 days. During the experiment, the appetite, hair, autonomous activity, and mental state of the mice were observed every day, and the body weight was weighed once every 3 days. Mice were sacrificed on day 28, and blood and liver samples were collected for subsequent analysis.

#### 2.4.4. Analysis of Physiological and Pathological Indexes

Viscera index: The experimental animal livers were weighed using an electronic balance, and the organ index was calculated according to the following formula:
Liver index (%) = liver weight (g)/mouse weight (g) × 100%

Hepatic function index: We precisely weighed out 0.1 g of liver tissue, washed it with normal saline, and placed it in a 10 mL centrifuge tube; then, 0.9 mL of normal saline was added. The mixture was homogenized by a homogenizer in an ice bath and centrifuged at 4 °C (3000 r·min^−1^, 10 min). The supernatant (10% liver homogenate) was obtained and numbered. Afterward, the enzymatic activities of ALT, AST, and ALP in the liver homogenate were measured using ELISA kits. Standard substances, samples, and horseradish peroxidase (HRP)-labeled detection antibodies were successively added to the precoated AST antibody-coated micropores, which were incubated and thoroughly washed. The color was developed using the substrate tetramethylbenzidine (TMB), which was converted to blue by catalase and then to yellow by acid. The depth of color was positively correlated with the AST concentration in the sample. The absorbance (OD) value was measured at 450 nm with a microplate reader, and the AST activity was calculated according to the standard curve. The same method was applied in the ALT and ALP operations.

Blood indicators: Blood was collected from the eye sockets of mice and placed into an ethylene diamine tetraacetic acid (EDTA) anticoagulant tube. Then, the white blood cells (WBCs), hemoglobin (HGB), mean hemoglobin content (MCH), mean hemoglobin concentration (MCHC), red blood cells (RBCs), hematocrit (HCT), mean red blood cell volume (MCV), and other indicators of the collected blood were analyzed using a GRT blood analyzer (MC: ANIMAL-6008, Jinan Grit Technology Co., Ltd., Jinan, China).

Histopathological examination: After the livers of the mice were fixed with 10% formaldehyde for 24 h, the same part of each group was cut into a size of about 1.0 × 1.0 × 0.2 cm^3^ and then dehydrated with gradient alcohol, xylene transparency, paraffin embedding, etc. After dehydration, the samples were embedded by an embedding machine to form a tissue wax block. Slices with a thickness of 5 μm were prepared and stained with hematoxylin–eosin (H & E). Subsequently, the histopathological changes were observed under an optical microscope.

### 2.5. Statistics and Analyses

SPSS 26.0 was used for the one-way analysis of variance and *t*-test of the data. The data are expressed as mean ± standard deviation ($\bar{x}$ ± SD). GraphPad Prism 8.0.2 was used for drawing graphs and SIMCA 14.1 was used for principal component analysis (PCA). A significant difference was set as *p* < 0.05, and an extremely significant difference was set as *p* < 0.01.

## 3. Results and Discussion

### 3.1. Impact on General Trait Indicators

In toxicological evaluation studies, general trait indicators such as appetite, hair, autonomous activity, and mental state are often used to visually evaluate the toxic reactions of various substances [29,30]. SPF male KM mice are male mice without specific pathogens that are native to Switzerland. Kunming mice account for 70% of all mice used in biomedical experiments in China due to their strong resistance to disease, strong adaptability, high reproductive rate, high survival rate, and relatively low price [31]. Moreover, Kunming mice have been found to have a higher correlation with humans, compared with ICR mice, suggesting that the Kunming mouse model may be more suitable for the study of liver injury [32]. Therefore, in this study, SPF male KM mice were selected as the main experimental animals with which to construct a cadmium-induced liver injury model. In this study, the daily behavior characteristics of mice were carefully observed, and the results showed that, throughout the whole study period, the appetite and autonomous activities of mice in the NC group were normal, their mental state was good, and their hair was shiny. Compared with the NC group, mice in the MC group showed anorexia, decreased voluntary activity, listlessness, and rough hair from the third day, and these pathological signs were significantly aggravated by the tenth day. (see Appendix A). These results suggest that cadmium exposure can significantly damage the appetite, autonomous activity, and mental state of mice, which was also confirmed in the J R Nation study [33].

Meanwhile, as shown in Figure 1A, the body weights of mice in the NC group showed a linear growth trend during the observation period of 21 days. Compared with the NC group, there was no significant difference in the initial weight gain of mice in the MC group. From the 7th day, the weight gain trend significantly slowed (*p* < 0.05), indicating that cadmium also had a certain impact on the growth of mice.

Polysaccharides are formed by the condensation and dehydration of multiple monosaccharide molecules. They are a type of carbohydrate substance with a complex and large molecular structure and are widely distributed in nature [34]. Some of them feature special biological activities; for example, heparin in the human body has an anticoagulant effect, and polysaccharides in pneumococcal cell walls have an antigenic effect [35]. ODP is one of the main active substances in *O. dillenii* (Ker-Gaw) Haw. In this study, we investigated the therapeutic effect of ODP on cadmium-exposed mice after different administration times at the same dosage (200 mg/kg). As shown in Appendix A, Compared with the NC group, mice in the ODP group showed a certain degree of anorexia, reduced autonomous activity, listlessness, and rough hair within 7 days of administration, but their overall situation was significantly better than that in the MC group. With the increase in the length of administration, the improvement in the appetite, autonomous activity, hair, and mental state of the ODP group mice became more obvious. By 28 days, the condition of these mice was similar to that of the NC group mice. In terms of the weight gain of mice, there was a significant difference in the changing trend after intervention with different lengths of administration, compared with that in the MC group. As shown in Figure 1B, and Appendix A, within 7 days of administration, the weight gain of mice in the MC, PC, and ODP groups was very similar without significant differences but was significantly lower than that in the NC group. By 14 days, the weight increment of mice in the PC and ODP groups did not change significantly, but both had significantly higher weights than those in the MC group. By 35 days, the weight gain of mice in the PC and ODP groups was very close to that in the NC group, while the weight gain trend of mice in the MC group was significantly lower than that in the PC and ODP groups. These results suggest that ODP has a potential protective effect on cadmium-exposed mice.

In this study, the changes in the property indexes of cadmium-exposed mice treated with different doses (50, 100, 200, 400, 600 mg/kg) of ODP were also investigated, and the results showed that the performance of mice treated with different doses of ODP differed within the 28-day observation period. Compared with the MC group, the mental state, hair shine, appetite, and autonomous activity of mice were improved to a certain extent at dosages of 50 and 100 mg/kg, but the improvement was significantly inferior to that of mice in the 200, 400, and 600 mg/kg groups. As for weight gain, as shown in Figure 1C, the weight gain of mice in the 50 and 100 mg/kg groups was not significantly different from that in the YC group but was significantly higher than that in the MC group, while the weight gain of mice in the 200, 400, and 600 mg/kg groups was significantly higher than that in the 50 and 100 mg/kg groups and showed a dose-dependent relationship.

### 3.2. Impact on Viscera Index

Normally, the ratio of organs to the body weight of mice is relatively constant. When mice are exposed to toxic substances, the weight of damaged organs could be changed, meaning that the organ coefficient is also changed. Therefore, the viscera index is often used to evaluate whether a particular substance is harmful to the body. In the present study, as shown in Figure 2A, the liver index of the MC group was significantly higher than that of the NC group (*p* < 0.01), revealing that cadmium exposure may cause serious damage to the liver.

Different durations of administration may impact the change in the liver index, so this study focused on the liver index of ODP administered for 1–5 weeks. As shown in Figure 2B, and Appendix A, the liver index of mice in the MC group and NC group was significantly different in different administration time ranges, while the liver index of mice in the YC and ODP groups was significantly lower than that in the MC group, indicating that ODP and YC can reduce the liver enlargement caused by cadmium exposure in mice.

The use of different doses of ODP may also affect the change in the liver index. Therefore, this study investigated the liver index after the administration of five different doses (50, 100, 200, 400, and 600 mg/kg) of ODP. As shown in Figure 2C, after treatment with different doses of ODP, compared with the MC group, the liver index of the mice in the administration group was significantly reduced, and the reduction rates were 12.64%, 5.05%, 12.36%, 18.48%, and 20.27%. This shows that ODP can reduce liver enlargement caused by cadmium exposure in mice, and there is a certain dependence on dosage.

### 3.3. Impact on Hepatic Function Index

The liver is a metabolic organ in the bodies of vertebrates and is rich in a variety of biological enzymes. Transaminase is the main enzyme in liver metabolism [36]. When transaminase activity is elevated, this is a sign that liver function has been damaged or destroyed [37]. AST and ALT are the two most important transaminases in the liver; therefore, the activities of AST and ALT are widely used to evaluate liver function damage [38]. In addition, ALP is found mainly in the liver and skeletal muscle, and an elevated ALP level indicates the presence of cholestasis in some diseases [39].

In the present study, changes in ALT, AST, and ALP were used to evaluate whether the liver function was normal or not after cadmium exposure. As shown in Table 1, compared with the NC group, the levels of ALT, AST, and ALP in the liver of cadmium-exposed mice were significantly higher (*p* < 0.01). The improvement rates were 26.61%, 45.61%, and 38.20%, respectively, indicating that the liver function was seriously impaired. Previous studies have proposed that cadmium can stimulate the levels of AST, ALT, and ALP, suggesting the presence of hepatocyte injury and necrosis is accompanied by intrahepatic cholestasis. In this study, our research conclusion is consistent with previous results.

A large number of studies have shown that polysaccharides have good protective and therapeutic effects against liver injury. For example, Jiang et al. [40] investigated the protective effect of oyster (Crassostrea gigas) polysaccharides on alcoholic liver injury, Hayaza et al. [41] reported the dual immunomodulatory effects of crude okra polysaccharide on carcinogenic liver injury in mice, and Bargougui et al. [42] revealed the protective effect of Stipa Parviflora Desf polysaccharide against CCl_4_-induced liver injury. However, there are still limited reports on the effects of polysaccharides against liver damage caused by heavy metals, especially cadmium, which is an important pollutant in the environment. Therefore, exploring natural substances with potential protective and therapeutic effects against cadmium-induced liver injury is of great significance for the research and development of relevant health foods and drugs.

In the present study, bifendate pills were selected as positive control agents and were used together with MC control mice to evaluate the anti-cadmium-induced liver injury efficacy of ODP after different lengths of administration. As shown in Table 2, after 1 week of administration, except that the ALT of mice in the MC group was significantly higher than that in the NC group (*p* < 0.05), the AST, ALT, and ALP levels in each group did not change significantly. By the second week, the levels of AST (*p* < 0.05), ALT (*p* < 0.05), and ALP (*p* < 0.01) in the MC group were significantly higher than those in the NC group, indicating that cadmium caused serious damage to the liver function of mice. At the same time, there was no significant difference in the AST, ALT, and ALP levels between mice in the ODP group and NC group, indicating that ODP played a protective or therapeutic role in cadmium-induced liver injury in mice. The situation in the third-to-fifth weeks was similar to that in the second week. The AST, ALT, and ALP levels in the MC group were significantly higher (*p* < 0.01) than those in the NC group, indicating that the damage caused by cadmium to liver function in the mice was continuing. Meanwhile, there was no significant difference in the levels of AST, ALT, and ALP between mice in the ODP group and the NC group, which indicated that ODP could play a continuous protective or therapeutic role in cadmium-induced liver injury in mice.

In this study, we also explored the protective or therapeutic effects of the use of different ODP doses on mice with cadmium-induced liver injury; the results are shown in Table 3. It can be seen from Table 3 that the levels of AST, ALT, and ALP in mice in the MC group were significantly higher than those in mice in the NC group (*p* < 0.01), which again proved that cadmium could lead to disordered AST, ALT, and ALP levels in mouse liver. After intervention with different doses of ODP (50, 100, 200, 400, and 600 mg/kg), the levels of AST, ALT, and ALP in the liver of mice were corrected, and there was no significant difference, compared with the NC group, indicating that the use of ODP in the five dose groups could restore the elevated levels of AST, ALT, and ALP induced by cadmium. In previous studies, it has been shown that the cyanobacteria Phormidim versicolor NCC466 polysaccharides and Periploca angustifolia polysaccharides can also significantly enhance liver function parameters (AST, ALT, and bilirubin), improve antioxidant capacity, and restore liver tissue changes caused by cadmium [43,44], which was consistent with our research results. These findings suggest that natural polysaccharides are potential sources of natural products possessing antioxidant, cytoprotective, and hepatoprotective properties.

### 3.4. Impact on Blood Indicators

Blood is a red opaque viscous fluid that flows in the blood vessels and hearts of humans or animals [45]. Blood contains a variety of nutrients, such as inorganic salt, oxygen, cell metabolites, hormones, enzymes, and antibodies. It has the function of nutritional tissue, regulating organ activity and preventing damage caused by harmful substances [46]. Hematological indexes play important roles in the diagnosis of many diseases; therefore, they are important evaluation metrics used in the discovery and research of functional substances [47]. In the present study, white blood cells (WBCs), hemoglobin (HGB), mean hemoglobin content (MCH), mean hemoglobin concentration (MCHC), red blood cells (RBCs), hematocrit (HCT), and mean red blood cell volume (MCV) were used to evaluate hematological changes.

HGB, MCH, and MCHC are associated with anemia. In general, if the levels of these three indicators are lower than normal, this suggests that iron deficiency anemia may occur. If MCH is greater than its normal value, and MCHC and HGB are slightly higher than their normal values, this may suggest anemia caused by vitamin B_12_ and folic acid deficiency. As shown in Figure 3A–C, the levels of HGB, MCH, and MCHC in mice after continuous exposure to cadmium were significantly lower than those in the blank group, which were reduced by 13.1%, 16.5%, and 8.0%, respectively, indicating that iron deficiency anemia may occur in mice after cadmium exposure. RBCs, MCV, and HCT have the ability to transport oxygen and carbon dioxide and buffer blood pH. As shown in Figure 3D–F, mice in the MC group showed no significant change in RBCs but revealed significant decreases in MCV and HCT, compared with mice in the NC group, suggesting that the heavy metal cadmium may have an effect on the transport of oxygen and carbon dioxide and the ability to buffer blood pH in mice. The main function of WBCs is defense, and different types of WBCs participate in the body’s defense response in different ways. If the number of WBCs in the body is higher than normal, it is likely that the body is suffering from inflammation. As shown in Figure 3G, the mean WBC count in the MC group was about 42.1% higher than that in the NC group, indicating that continuous exposure to cadmium may lead to severe inflammation.

From the above analysis, it can be seen that continuous exposure to cadmium may cause disorders in many blood indexes. Therefore, it is necessary to find relevant interventions and therapeutic substances. In the present study, we also investigated the effects of ODP on hematological parameters in mice continuously exposed to cadmium for different durations of intervention.

As shown in Figure 4 and Appendix A, after 7 days of cadmium exposure, compared with the NC group, the HGB, HCT, WBCs, MCH, MCHC, RBCs, and MCV of the ODP group showed no significant change, suggesting that ODP can correct the HGB, HCT, and WBC abnormalities caused by cadmium exposure. After 14 days of cadmium exposure, compared with the NC group, the seven hematological indexes in the MC group showed changes (*p* < 0.05). Among these, the levels of HGB, MCH, MCHC, RBCs, MCV, and HCT showed significant downward trends, and WBCs showed a significant upward trend. However, after ODP intervention, HGB, MCH, MCHC, RBCs, MCV, and HCT of the ODP group were not statistically different from those of the NC group, revealing that these abnormal hematologic markers had been well repaired. Within 14–35 days of cadmium exposure, the change characteristics of hematological indexes in mice were very similar to those observed at 2 weeks. Multiple indexes in the model group were significantly increased or decreased, compared with those in the blank group, while those in the ODP intervention group were basically unchanged, showing that the normal state of hematological indexes had been well maintained.

In order to further study the influence of ODP on the hematology of mice after cadmium exposure, we analyzed the changing trends of HGB, HCT, WBCs, MCH, MCHC, RBCs, and MCV subjected to intervention at doses of 50, 100, 200, 400 and 600 mg/kg; the results are shown in Figure 5. The dose relationship showed that the contents of RBCs, HCT, and MCV in each administration group were increased, compared with those in the MC group. A dose of 50 mg/kg of ODP significantly increased the RBC and HCT levels; a dose of 100 mg/kg of ODP significantly increased the HCT levels; a dose of 200 mg/kg of ODP significantly increased the RBC, HCT, and MCV levels; a dose of 400 mg/kg of ODP significantly increased the HCT and MCV levels; and a dose of 600 mg/kg of ODP significantly increased the RBC and HCT levels (*p* < 0.05).

### 3.5. Impact on Pathological Morphology

Pathological examination is an important basis for judging the degree of liver injury [48]. In this study, we observed the pathological morphology of mouse liver exposed to cadmium for different lengths of time; the results are shown in Figure 6. The liver cell structures of the NC group (Figure 6A–E) were complete, the nucleus was clearly visible, and the cytoplasm showed no histological abnormalities. 

Hepatocyte swelling, nuclear exposure, central venous congestion, apoptosis, and inflammatory cell infiltration were observed in the MC group (Figure 6F–J), and comparisons of pathological sections of liver cells in all the ODP groups (Figure 6P–T and Figure 7) and the MC group were also made. ODP was administered 14 days later; the pathological changes in liver cells were somewhat relieved, but most of the cell structures were still deformed (Q). With the increase in the duration of the administration period, the pathological changes in hepatocytes were gradually alleviated (Q-T). Intriguingly, *O. dillenii* extract has been shown to be able to alleviate sinusoidal dilatation, central venous congestion, vacuolation, and inflammatory cell infiltration in the liver and alleviate pathological changes in the liver [49]. In addition, Agaricus blazei Murill polysaccharides [50] have been shown to be able to reduce histopathological damage, improve liver cell lesions, and restore normal liver cell morphology, providing confirmation of the efficacy of the use of polysaccharides against liver injury.

### 3.6. Multivariate Statistics

In this study, principal component analysis (PCA) analysis was carried out on the peripheral blood of mice administered ODP for 1–5 weeks, and the overall distribution trend of samples during this period was observed. A PCA score chart is shown in Figure 8A and Appendix A. The NC and MC groups showed a specific separation trend, and these two groups of samples could be separated well. The YC and ODP groups were close to the NC group on the PCA score map, indicating that ODP and bifendate have good protective effects on the liver and can delay cadmium-induced liver injury for different lengths of time. Partial least squares discriminant analysis (OPLS-DA) can filter out noises irrelevant to classification information and improve the analytical ability and effectiveness of a model [51]. The OPLS-DA score of peripheral blood after 5 weeks of ODP administration is shown in Figure 8B, which showed that the OPLS-DA model could distinguish the NC group from the MC group well. To avoid the overfitting of OPLS-DA in the modeling process, a replacement test was carried out on the model to ensure its validity. Figure 8C shows the model replacement inspection chart obtained after 200 cycles of interactive verification. The Q^2^ was 0.0126, and the R^2^ was −0.0344. Therefore, it can be observed that the OPLS-DA model was not overfitted, and its prediction results were stable and reliable.

The PCA scores obtained for the oral administration of ODP for 5 weeks are shown in Figure 8D–F. The results show that the differences between 2, 3, and 5 weeks of administration were similar, while the difference between week 1 and week 4 was significant, which showed that there was a good separation trend between 1 week and 4 weeks. Then, a PCA analysis was carried out on the blood routine data obtained for the NC, MC, YC, and ODPC groups at 1 week and 4 weeks. The results showed that the ODPC and MC groups had no good separation tendency 7 days after administration Figure 8G–I, but the ODPC and MC groups had a good separation tendency at 28 days Figure 8J–L, and the ODPC group had a similar tendency to the NC group, which indicated that an administration duration of 28 days was the best in terms of ODP efficacy.

## 4. Conclusions

In summary, exposure to cadmium can lead to slow weight gain; liver enlargement; increased AST, ALT, and ALP levels; and decreased HGB, MCH, and MCHC concentrations. In addition, it also causes serious pathological changes. ODP has an obvious protective effect on early, middle, and late liver injury caused by cadmium exposure. The onset time occurs within 7 days after administration, the efficacy reaches the highest level after continuous administration for 14 days, and this trend can continue until the time point of 35 days. Different doses of ODP show a good dose–effect relationship with cadmium-induced liver injury. These findings indicate that ODP had excellent potential for use as a health food or therapeutic drug.

## Figures and Tables

**Figure 1 foods-11-01340-f001:**
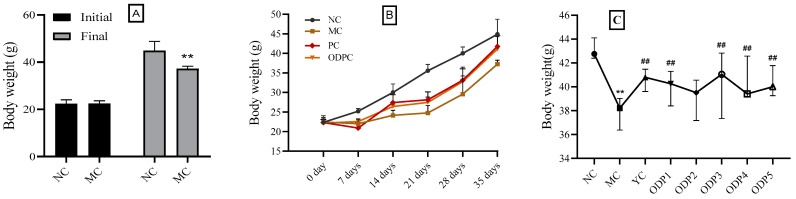
Changes in body weight gain in different groups of mice: (**A**) effects of cadmium exposure on the body weight of mice; (**B**) time–effect study of ODP; (**C**) dose–effect study of ODP. NC: normal control group, MC: CdCl_2_-treated group, YC: positive-treated group, ODPC: ODP-treated group. ODP1–5: ODP (50, 100, 200, 400, 600 mg/kg)-treated group. The presented values are x ± SD (*n* = 6 for each group). * Compared with normal control group (NC group), ** *p* < 0.01; ^#^ compared with model control group (MC group), ^##^ *p* < 0.01.

**Figure 2 foods-11-01340-f002:**
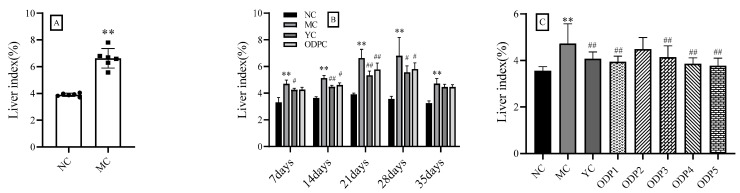
Changes in liver index in different groups of mice: (**A**) effects of cadmium exposure on the liver index of mice; (**B**) time–effect study of ODP; (**C**) dose–effect study of ODP. NC: normal control group, MC: CdCl_2_-treated group, YC: positive-treated group, ODPC: ODP-treated group, ODP1–5: ODP (50, 100, 200, 400, 600 mg/kg)-treated group. The presented values are x ± SD (*n* = 6 for each group). * Compared with normal control group (NC group), ** *p* < 0.01; ^#^ compared with model control group (MC group), ^#^ *p* < 0.05, ^##^ *p* < 0.01.

**Figure 3 foods-11-01340-f003:**
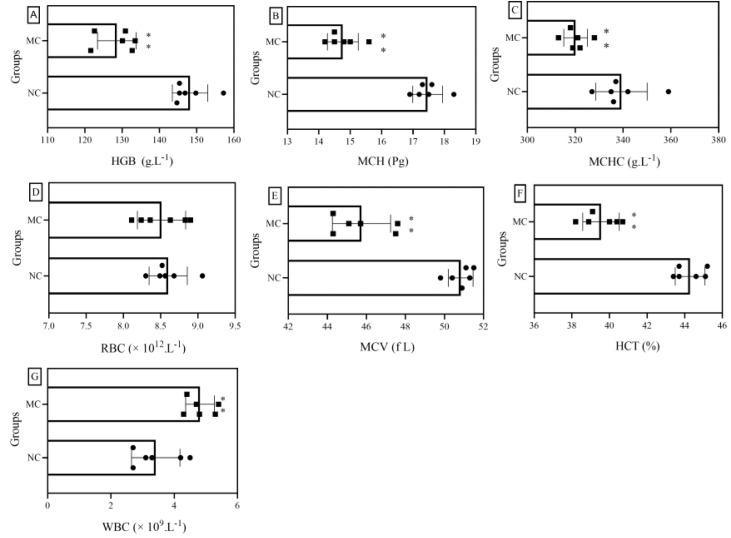
Effects of cadmium exposure on hematological indicators of mice: (**A**–**G**) are HGB, MCH, MCHC, RBCs, MCV, HCT, and WBCs, respectively. NC: normal control group, MC: CdCl_2_-treated group. * Compared with normal control group (NC group), **, *p* < 0.01.

**Figure 4 foods-11-01340-f004:**
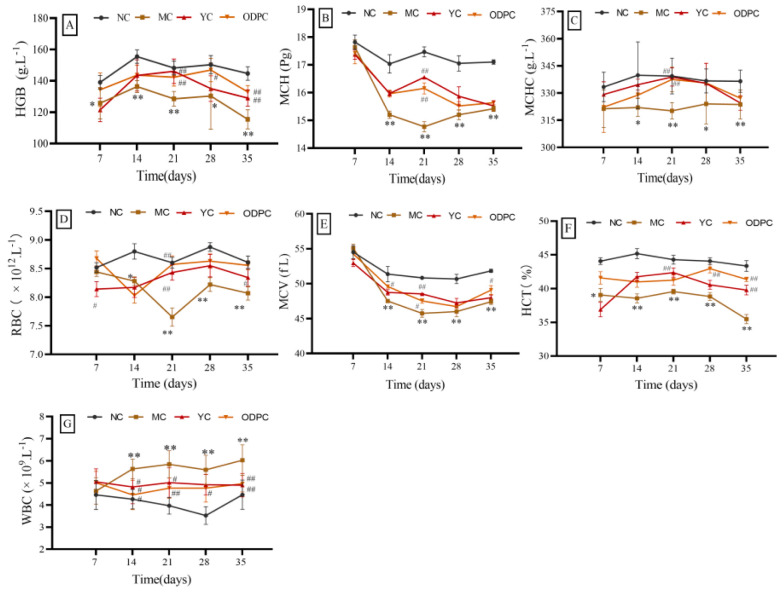
Time–effect relationship of ODP on peripheral blood indexes in mice: (**A**–**G**) are HGB, MCH, MCHC, RBCs, MCV, HCT, and WBCs, respectively. NC: normal control group, MC: CdCl_2_-treated group, YC: positive-treated group, ODPC: ODP-treated group. * Compared with normal control group (NC group), # compared with model control group (MC group). * *p* < 0.05, ** *p* < 0.01; ^#^ *p* < 0.05, ^##^ *p* < 0.01.

**Figure 5 foods-11-01340-f005:**
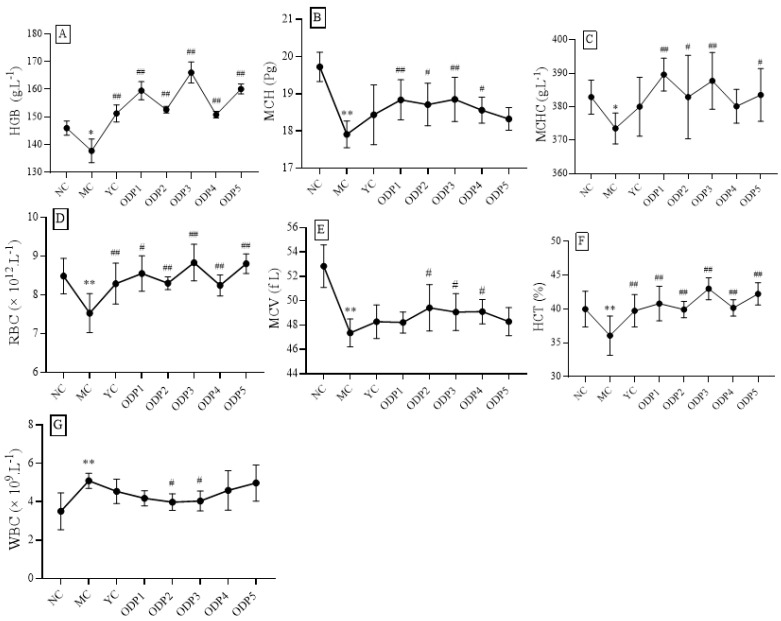
Dose–response relationship of ODP in mice peripheral blood: (**A**–**G**) are HGB, MCH, MCHC, RBCs, MCV, HCT, and WBCs, respectively. NC: normal control group, MC: CdCl_2_-treated group, YC: positive-treated group, ODP1–5: ODP (50, 100, 200, 400, 600 mg/kg)-treated group. The presented values are x ± SD (*n* = 6 for each group). Compared with normal control group (NC group), # compared with model control group (MC group). * *p* < 0.05, ** *p* < 0.01; ^#^ *p* < 0.05, ^##^ *p* < 0.01.

**Figure 6 foods-11-01340-f006:**
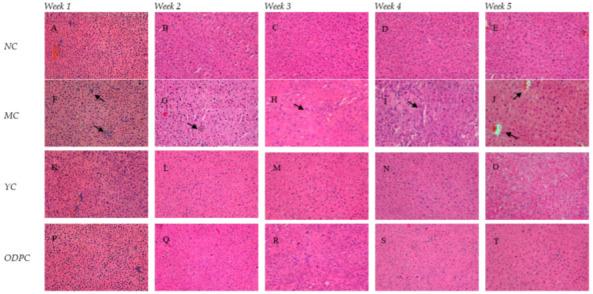
Representative H & E micrographs of time-dependent liver histopathology (400×): (**A**–**E**) NC group, (**F**–**J**) MC group, (**K**–**O**) YC group, and (**P**–**T**) ODPC group. NC: normal control group, MC: CdCl_2_-treated group, YC: positive-treated group, ODPC: ODP-treated group.

**Figure 7 foods-11-01340-f007:**
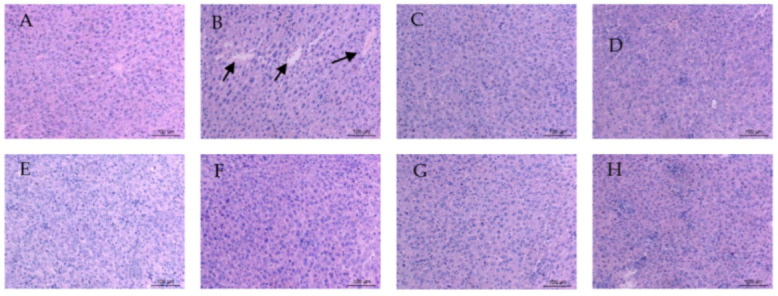
Representative H & E micrographs of dose–response liver histopathology (400×): (**A**) NC group; (**B**) MC group; (**C**) YC group; (**D**–**H**) ODP 1–5 group. NC: normal control group, MC: CdCl_2_-treated group, YC: positive-treated group. ODP1–5: ODP (50, 100, 200, 400, 600 mg/kg)-treated group.

**Figure 8 foods-11-01340-f008:**
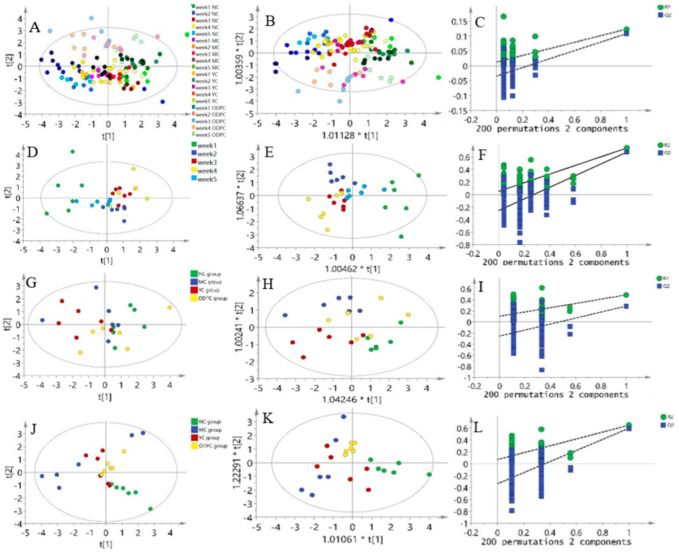
Results of the multivariate statistical analysis of the ODP time dependence study: (**A**) PCA score chart of peripheral blood; (**B**) OPLS-DA score plot of peripheral blood; (**C**) OPLS-DA replacement test diagram of peripheral blood; (**D**) PCA score chart of oral ODP for 35 days; (**E**) OPLS-DA score chart of oral ODP for 35 days; (**F**) OPLS-DA replacement test chart of oral ODP for 35 days; (**G**) PCA score chart of oral ODP for 7 days; (**H**) OPLS-DA score chart of oral ODP for 7 days; (**I**) OPLS-DA replacement test chart of oral ODP for 7 days; (**J**) PCA score chart of oral ODP for 28 days; (**K**) OPLS-DA score chart of oral ODP for 28 days; (**L**) OPLS-DA replacement test chart of oral ODP for 28 days. NC: normal control group. MC: CdCl_2_-treated group. YC: positive-treated group. ODPC: ODP-treated group.

**Table 1 foods-11-01340-t001:** Effects of cadmium exposure on AST, ALT, and ALP in the liver of mice.

Group/Index	AST(U/L)	ALT(U/L)	ALP(U/L)
NC	48.56 ± 5.36	46.24 ± 4.36	4.79 ± 0.39
MC	61.48 ± 6.32 **	67.33 ± 7.41 **	6.62 ± 0.47 **

NC: normal control group, MC: CdCl_2_-treated group. * Compared with normal control group (NC group),** *p* < 0.01.

**Table 2 foods-11-01340-t002:** Time–effect relationship of ODP on AST, ALT, and ALP in the liver of mice.

	Group/Time	Week 1	Week 2	Week 3	Week 4	Week 5
AST (U/L)	NC	42.51 ± 3.78	42.80 ± 1.52	48.56 ± 5.36	44.97 ± 6.99	47.94 ± 3.98
	MC	47.07 ± 5.16	49.31 ± 6.31 *	61.48 ± 6.32 **	53.87 ± 8.98 **	57.20 ± 3.61 **
	YC	45.45 ± 4.75	45.14 ± 4.64	58.05 ± 6.46 ^#^	50.73 ± 5.96 ^#^	48.63 ± 4.85 ^##^
	ODPC	45.07 ± 3.57	42.57 ± 6.91	57.28 ± 4.19 ^#^	45.81 ± 2.98 ^##^	49.49 ± 5.99 ^##^
ALT (U/L)	NC	47.45 ± 3.2	50.21 ± 3.36	46.24 ± 4.36	44.18 ± 3.01	45.92 ± 2.62
	MC	53.76 ± 2.19 *	56.04 ± 8.91 *	67.33 ± 7.41 **	62.53 ± 4.13 **	63.84 ± 5.73 **
	YC	50.91 ± 4.93	52.35 ± 6.33	52.73 ± 6.44 ^#^	52.93 ± 2.23 ^#^	49.74 ± 6.86 ^##^
	ODPC	50.49 ± 7.91	52.07 ± 5.43	51.45 ± 8.80 ^#^	45.58 ± 6.85 ^##^	47.17 ± 3.82 ^##^
ALP (U/L)	NC	5.22 ± 0.64	5.1 ± 0.38	4.79 ± 0.39	5.55 ± 0.25	4.80 ± 0.55
	MC	5.50 ± 1.42	6.13 ± 0.76 **	6.62 ± 0.47 **	6.74 ± 1.19 **	6.38 ± 0.52 **
	YC	5.37 ± 1.05	5.30 ± 0.55 ^#^	5.62 ± 0.64 ^#^	5.85 ± 0.31 ^#^	5.57 ± 0.58 ^#^
	ODPC	5.30 ± 0.91	5.34 ± 0.31 ^#^	5.55 ± 0.92 ^#^	5.40 ± 0.40 ^##^	5.02 ± 0.33 ^##^

NC: normal control group, MC: CdCl_2_-treated group, YC: positive-treated group, ODPC: ODP-treated group. * Compared with normal control group (NC group), ^#^ compared with model control group (MC group). * *p* < 0.05, ** *p* < 0.01; ^#^ *p* < 0.05, ^##^
*p* < 0.01.

**Table 3 foods-11-01340-t003:** Dose–response relationship of ODP with AST, ALT, and ALP in mice.

Group/Index	AST(U/L)	ALT(U/L)	ALP(U/L)
NC	38.46 ± 6.65	71.88 ± 6.72	5.67 ± 1.00
MC	57.7 ± 3.38 **	101.29 ± 12.25 **	9.09 ± 1.54 **
YC	41.52 ± 5.46 ^##^	76.59 ± 8.11 ^#^	6.52 ± 1.09 ^##^
ODP1 (50 mg/kg)	46.26 ± 5.82 ^##^	86.23 ± 6.03	7.63 ± 0.91 ^#^
ODP2 (100 mg/kg)	46.93 ± 9.31 ^##^	85.90 ± 10.94	7.64 ± 0.88 ^#^
ODP3 (200 mg/kg)	43.06 ± 4.31 ^##^	71.55 ± 5.95 ^##^	6.96 ± 0.50 ^##^
ODP4 (400 mg/kg)	44.10 ± 5.02 ^##^	71.85 ± 8.24 ^##^	6.60 ± 0.51 ^##^
ODP5 (600 mg/kg)	50.41 ± 2.18 ^##^	81.34 ± 15.41	6.85 ± 1.31 ^##^

NC: normal control group, MC: CdCl_2_-treated group. YC: positive-treated group. ODP1–5: ODP (50, 100, 200, 400, 600 mg/kg)-treated group. * Compared with normal control group (NC group), ** *p* < 0.01; ^#^ compared with model control group (MC group), ^#^ *p* < 0.05, ^##^ *p* < 0.01.

## Data Availability

The data presented in this study are available on request from the corresponding author.

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
