# Peer review of "A Study on the Time–Effect and Dose–Effect Relationships of Polysaccharide from Opuntia dillenii against Cadmium-Induced Liver Injury in Mice"

_foods, 2022, doi:10.3390/foods11091340_

Round 1

Reviewer 1 Report

General Comment

The authors investigated the time and dose effects of polysaccharide from Opuntia dillenii against cadmium-induced liver injury. Generally, the manuscript was well written, use sufficient number of animals and well analyzed data. However, some issues needed to be taken care of. Find below, the areas.

Abstract

  1. The abstract should be a continuous- one paragraph.
  2. All the abbreviations should be written in full at first mentioning
  3. Some grammatical errors were noticed; for instance, …. ODP was applied to intervention the mice….

Introduction

  1. It will be good to cite references of some chemical drugs used to treat cadmium-induced liver injury but also cause secondary liver damage’
  2. At times, what the author is imply is unclear. For instance, …. For example, glycyrrhiza and salidroside could prevent cadmium induced cell death ("Cytoprotective effects of Glycyrrhizae radix extract and its active component liquiritigenin against cadmium-induced toxicity (effects on bad translocation and cytochrome c-mediated PARP cleavage)," 2004;
    Hui et al., 2014) … this area is unclear.
  3. Change protected to protect in this sentence ….’’reduce the accumulation of cadmium
    in the liver and protected liver cells from damage’’
  4. Indeed, the hepatoprotective effect of dilleni has been well explored in rat model (DOI: 10.4103/2221-1691.233006), mice (Shah et al., 2016), and DOI:10.4103/2221-1691.233006, just to mention few with even the biochemical assays investigated already explored.  Authors should please point out the areas of novelty in the manuscript.

Method

  1. How was the sample size determined?
  2. A reference is needed to substantiate the induction of liver injury using cadmium chloride.
  3. Indicate that the ALP, ALT, and AST kits used were ELISA; then briefly describe how the assays were done.
  4. How was the dose of ODPC used determined?
  5. What was used to dissolve the ODPC?

Results and Discussion

  • The claim that cadmium could decrease appetite in mice requires citation.
  • Why did author choose to plot different kinds of graphs for the 3 models? The results could not be properly followed; thus, it is difficult to identify trends and lack of it.
  • Each figure legend should be clearly labelled regardless whether the acronym has been used before and/or elsewhere. Author should state the meaning of NC, YC, and on under each graph (figure 1).
  • Because the results and discussion were combined, the biochemical mechanism underlying the ODC hepatoprotective effect could not be really discussed.
  • Again, the graphs, even though were used to describe similar parameters could be used consistently. For instance, the graph C is concealing the errors in the measurement here. (Figure 2).
  • The discussion section looks more descriptive and did not discuss the results obtained to details.
  • Figure 5 is difficult to understand. Author should consistently use graphs that will be understandable.
  • The author used different in-citing method while discussing the pathological aspect. Use one format consistently.
  • The PCA analyses is meant to simplify the somewhat complex data. The author did not harmonize all the results obtained to substantiate the claim well. It will be profitable if the discussion be separated from the results interpretation as this aspect looks clumsy.

Reviewer 2 Report

  1. Opuntia dillenii should be in italic throughout the manuscript including the title.
  2. The objective of the work should be clearly spelt clearly. Polysaccharides possessing good antioxidant activity should not be the only reason.
  3. Why SPF male KM mice were used for the study? Need to describe with reference in discussion
  4. What is the reason to arrive at the concentration of ODP as 50, 100, 200, 400, 600 mg / kg?
  5. The status of other polysaccharides isolated from other species in the same study should be compared and discussed
  6. Figure 7 and 8 image quality can be enhanced
  7. The authors have not mentioned the essential information on the polysaccharides physicochemical characteristics like molecular weight, polydispersity, intrinsic viscosity etc. This will contribute significantly to the improvement of the manuscript
  8. Requires English edits
